# *PTPN2* Regulates Iron Handling Protein Expression in Inflammatory Bowel Disease Patients and Prevents Iron Deficiency in Mice

**DOI:** 10.3390/ijms26073356

**Published:** 2025-04-03

**Authors:** Hillmin Lei, Ali Shawki, Alina N. Santos, Vinicius Canale, Salomon Manz, Meli’sa S. Crawford, Pritha Chatterjee, Marianne R. Spalinger, Michael Scharl, Declan F. McCole

**Affiliations:** 1School of Medicine, Division of Biomedical Sciences, University of California, Riverside, CA 92521, USA; hlei013@ucr.edu (H.L.);; 2Department of Gastroenterology & Hepatology, University Hospital Zurich, 8091 Zurich, Switzerland

**Keywords:** inflammatory bowel disease, anemia, iron deficiency

## Abstract

Anemia is the most common extraintestinal manifestation of inflammatory bowel disease (IBD). Iron deficiency is the most frequent cause of anemia in IBD; however, the mechanisms involved are still poorly understood. Here, we investigated the role of the IBD risk gene, protein tyrosine phosphatase non-receptor type 2 (*PTPN2*), in regulating iron homeostasis. Proteomic analyses were performed on serum from IBD patients genotyped for the IBD-associated loss-of-function rs1893217 *PTPN2* variant. Constitutive *Ptpn2* wild type (WT), heterozygous (Het), and knockout (KO) mice were analyzed for iron content, blood parameters, and expression of iron handling proteins. Iron absorption was assessed through radiotracer assays. Serum proteomic analyses revealed that the “iron homeostasis signaling pathway” was the main pathway downregulated in Crohn’s disease (CD) patients carrying the *PTPN2* risk allele, independent of disease activity. *Ptpn2*-KO mice showed characteristics of anemia, including reduced hemoglobin concentrations along with serum and tissue iron deficiency and elevated serum hepcidin levels vs. *Ptpn2*-WT and Het mice. ^55^Fe absorption via oral gavage was significantly impaired in *Ptpn2*-KO mice. Correspondingly, *Ptpn2*-KO mice showed reduced apical membrane expression of the iron transporter DMT1. CD patients with the *PTPN2* loss-of-function rs1893217 variant display alterations in serum iron handling proteins. Loss of *Ptpn2* in mice caused features of anemia, including iron deficiency associated with reduced apical membrane expression of DMT1. These findings identify an important role for *PTPN2* in regulating systemic iron homeostasis.

## 1. Introduction

The inflammatory bowel diseases (IBDs), Crohn’s disease (CD) and ulcerative colitis (UC), are chronic inflammatory disorders of the gastrointestinal tract, which affect more than seven million people worldwide [1]. Anemia, a condition characterized by a reduction of functional erythrocytes, is the most frequent extraintestinal manifestation of IBD and significantly reduces the quality of life while increasing both the hospitalization rates and medical care costs of IBD [2,3]. The prevalence of anemia varies from 13% to 90% due to differences in the definition and patient groups examined, with a higher prevalence in hospitalized IBD patients than outpatients [4]. Anemic IBD patients can experience additional symptoms including dyspnea, tachycardia, fatigue, and reduced cognitive function, which can further increase morbidity rates compared to non-anemic IBD patients [5].

Iron deficiency is the most common cause of anemia in IBD, and insufficient iron levels can still significantly impair the health of IBD patients [2,6]. The two major etiologies of iron deficiency anemia are true iron deficiency anemia (IDA) and anemia of chronic disease (ACD), which are the most common causes of anemia worldwide [2]. Moreover, anemia in IBD is most likely multifactorial in origin and features a combination of IDA and ACD, which further complicates the manifestation of the condition [2]. Reliable diagnosis of the type and cause of anemia is also challenging due to the lack of a specific biomarker for the determination of iron deficiency in inflammation [4,7]. Although the current guidelines recommend screening and treatment of anemia in IBD patients, it remains underdiagnosed and undertreated by gastroenterologists [4,7]. Therefore, understanding the mechanisms of anemia in IBD is essential to develop effective IBD therapeutics.

Despite a longstanding awareness of the prevalence of anemia in IBD, the mechanisms of iron deficiency in IBD are still poorly understood. While there is currently no known physiological regulatory mechanism for the excretion of iron, iron loss can occur through indirect and non-regulated mechanisms, including cell desquamation or bleeding [8]. Maintenance of overall body iron levels occurs primarily through enterocyte absorption of iron in the proximal small intestine while reticuloendothelial macrophages of the liver, spleen, and bone marrow recycle iron from senescent erythrocytes [8]. Furthermore, regulation of enterocyte iron uptake and export of iron stores from macrophages and hepatocytes is tightly controlled by the iron regulatory hormone hepcidin anti-microbial peptide 1 (HAMP1) through its degradation of the cellular transmembrane iron export protein, ferroportin (FPN) [8]. As iron is an essential component of heme and hemoglobin, iron malabsorption, continuous iron loss, or iron dysregulation may limit iron uptake in erythrocyte precursors and subsequently restrict hemoglobin synthesis and erythropoiesis thereby leading to anemia [8]. In nonerythroid cells, altered synthesis of iron handling proteins may affect the maintenance of epithelia undergoing rapid turnover [9]. Interestingly, ACD in IBD positively correlates with the levels of interleukin-6 (IL-6), a pro-inflammatory cytokine that is transiently upregulated in chronic inflammatory diseases [10]. IL-6 may also suppress erythropoiesis by reduction of hemoglobin levels and can significantly upregulate hepcidin expression via the Janus kinase 1 (JAK1)-signal transducer and activator of the transcription 3 (STAT3) pathway [8,11].

A key negative regulator of JAK1–STAT3 pathway activation is protein tyrosine phosphatase non-receptor type 2 (PTPN2) [12,13]. Loss-of-function (LOF) mutations caused by single nucleotide polymorphisms (SNPs) in *PTPN2* increase the risk for IBD [14,15,16]. Thus, determining the functional consequences of *PTPN2* SNPs on iron transport may help identify the role of *PTPN2* in regulating disease-relevant biological pathways, as well as future treatments to alleviate extraintestinal complications in IBD. Here, we describe the identification of the “iron homeostasis signaling pathway” as the primary pathway downregulated in serum samples from CD patients carrying a *PTPN2* LOF SNP, rs1893217. Using in vivo models, we further identify a critical role for the IBD candidate gene, *PTPN2*, in iron homeostasis.

## 2. Results

### 2.1. PTPN2 rs1893217 Risk Allele Alters Serum Iron Handling Proteins in IBD Patients

In collaboration with the Swiss IBD Cohort (SIBDC) and the City of Hope Translational Biomarker Discovery Core Facility, we performed a proteomics screen on serum samples from IBD patients genotyped with the IBD-associated loss-of-function (LOF) rs1893217 SNP in *PTPN2* [14,15,16]. CD and UC patients (48% female) were categorized according to disease activity and the presence of the homozygous risk allele (CC), heterozygous patients carrying one copy of the risk allele (CT), and patients with wild-type *PTPN2* lacking the risk allele (TT) (Table 1A–C). Serum proteomic analyses identified the “iron homeostasis signaling pathway” as the primary pathway downregulated in CD but not UC patients, with the risk allele independent of disease activity (Figure 1). Downregulation of the major serum iron carrier protein, transferrin (TF), was observed in IBD patients with the rs1893217 SNP (Figure 1A). Notably, serum transferrin receptor 1 (TFR1) levels were reduced in CD patients (Figure 1A). Interestingly, the serum heme-binding protein hemopexin (HPX), which recovers unbound heme, was downregulated in UC patients but upregulated in CD patients (Figure 1A,B), indicating differential regulation of this protein in IBD subtypes. Upregulation of haptoglobin (HP) and haptoglobin-related protein (HPR), both of which bind free hemoglobin for hepatic heme iron recycling, occurred in UC but not CD, and this may represent a compensatory response to selectively minimize iron deficiency in UC but not CD (Figure 1B). It is notable that inflammatory activity—categorized as disease severity—had quite different effects on iron homeostasis proteins from the presence of the *PTPN2* gene variant. Not only were a distinct set of iron handling proteins altered by the disease severity, but the overall directionality of the effects was the opposite of those altered by the *PTPN2* variant, with iron regulatory proteins largely increased in Crohn’s disease patients. Alpha-2-macroglobulin (A2M) and HP were downregulated in CD patients while RAB7A was downregulated in both CD and UC patients (Figure 1C). Apolipoprotein E (APOE) was downregulated only in UC patients (Figure 1C). In contrast, hemoglobin proteins (HBA1, HBA2, HBB, HBG1, HBG2, HBE1, and HBD) were upregulated in CD patients (Figure 1D). Marginally more CD patients with the “C” risk allele had anemia (hemoglobin concentrations < 12.0 or <13.0 g/dL in females and males, respectively) than patients with the WT alleles (20% vs. 11%), despite fewer “C” risk allele CD patients having severe active disease (Table 1), although this did not reach statistical significance. These data indicate a major role for *PTPN2* in regulating iron handling genes in IBD and identify regulatory networks that are substantially altered by the rs1893217 SNP independent from effects on iron regulatory proteins associated with inflammatory activity.

### 2.2. Constitutive Loss of Ptpn2 in Mice Causes Features of Anemia Including Iron Deficiency

To determine if *Ptpn2* deficiency in mice could be utilized as an experimental model to study iron depletion in humans with LOF *PTPN2* variants, it was essential to identify whether these mice exhibit any defects in iron levels. Quantification of serum iron showed that *Ptpn2*-KO mice have reduced serum iron levels and TF saturation when compared with wild-type (WT) and Het mice (Figure 2A,B) [17]. Furthermore, automated CBC revealed that these *Ptpn2*-KO mice have reduced hematocrit levels and hemoglobin concentrations, suggesting the presence of anemia (Figure 2C,D) [18,19]. Regarding the type of anemia present, the mean corpuscular volume (MCV) of erythrocytes was decreased in *Ptpn2*-deficient mice, consistent with the features of IDA compared to ACD where MCV should be unaltered (Figure 2E) [19]. Notably, hemoccult analysis did not show any detectable blood in the stool of *Ptpn2*-KO vs. WT or Het mice (n = 5), demonstrating that the iron deficiency observed in these mice is likely not due to blood loss. Collectively, these data indicate that *Ptpn2*-deficient mice have features of anemia and can be utilized as an experimental model to study mechanisms of iron deficiency upon loss of (functional) PTPN2.

### 2.3. Ptpn2-KO Mice Have Reduced Iron Storage and Impaired Small Intestinal Iron Homeostasis

Given the marked defects in serum iron levels and features of anemia we observed in whole-body *Ptpn2*-deficient mice, we investigated whether *Ptpn2*-KO mice have diminished tissue iron levels. Quantification and analysis of non-heme iron content, including intracellular iron and iron stored in the iron storage molecule ferritin (FT), showed reduced non-heme iron levels in several extraintestinal tissues including the liver, spleen, kidney, and gastrocnemius muscle of *Ptpn2*-KO mice compared to *Ptpn2*-WT and Het mice, suggesting lower tissue storage of iron (Figure 3A–D). Moreover, *Ptpn2*-KO mice have significant splenomegaly [20,21,22]. We then assessed intestinal non-heme iron levels, as iron homeostasis is mainly controlled by small intestinal absorption [23]. *Ptpn2*-KO mice showed a substantial reduction in duodenal and jejunal non-heme iron levels, indicating that iron handling in the proximal small intestine is compromised (Figure 3E,F). No significant changes in non-heme iron levels were seen in the ileum, cecum, proximal colon, or distal colon intestinal segments of *Ptpn2*-deficient vs. WT or Het mice (Figure 3G–J). These results suggest that *Ptpn2*-KO mice have defects in body iron stores and altered iron homeostasis in the proximal small intestine, the major site of iron absorption.

### 2.4. Constitutive Ptpn2 Deficiency in Mice Reduces Duodenal Iron Absorption

To examine whether constitutive *Ptpn2* deficiency alters iron absorption, we measured radiotracer iron absorption in *Ptpn2*-KO mice vs. WT and Het mice 2 h after oral-intragastric gavage of a ^55^Fe dose. In whole-body *Ptpn2*-deficient mice, ^55^Fe content in duodenal IECs was significantly reduced compared to *Ptpn2*-WT and Het mice 2 h after gavage (Figure 4A). The lack of ^55^Fe present in duodenal IECs of *Ptpn2*-KO mice provides evidence that a contributor to the iron deficiency apparent in these mice might be a defect at the duodenal brush border. Similarly, the liver is expected to rapidly clear ^55^Fe from the portal and peripheral circulation; however, liver ^55^Fe content remained low in *Ptpn2*-deficient mice (Figure 4B) [24,25]. These data suggest that constitutive *Ptpn2* loss results in reduced iron absorption.

### 2.5. Ptpn2-KO Mouse Duodenal Enterocytes Show Alterations in Iron Handling Proteins

To identify the mechanism(s) by which whole-body *Ptpn2*-deficient mice show impaired iron absorption, we collected duodenal IECs to analyze iron regulatory gene and protein expression [8]. We found that the expression of iron uptake genes, including the intestinal ferrireductase cytochrome b reductase 1 (*Cybrd1*), the brush border ferrous iron transporter divalent metal transport 1 (*Dmt1*), and the iron absorption driver, sodium-hydrogen exchanger 3 (*Nhe3*), was unaltered in duodenal IECs from *Ptpn2*-KO mice through quantitative polymerase chain reaction (qPCR) (Figure 5A–C). Western blot analysis of duodenal IECs revealed a significant reduction in the protein levels of the iron transporter DMT1, while NHE3 levels were unaltered (Figure 5D,E). To confirm the reduction of DMT1 in duodenal IECs of *Ptpn2*-KO mice, we performed immunofluorescence (IF) staining for DMT1 on duodenal sections. IF images revealed that the duodenal epithelium of *Ptpn2*-deficient mice showed reduced apical membrane expression of DMT1 compared to *Ptpn2*-WT and Het mice, suggesting a possible mechanism of impaired intestinal non-heme iron uptake (Figure 5F). Furthermore, *Ptpn2*-KO mice showed unaltered NHE3 membrane localization (Figure 5G), demonstrating that iron absorption is impeded by the reduced levels of DMT1 on the intestinal brush border rather than its collaborating transporter, NHE3. In addition, qPCR and Western blot analysis also revealed reduced expression of the iron storage molecule FT—detected by ferritin heavy chain 1 (*Fth1*)—(Figure 6A,C), while expression of the basolateral iron importer transferrin receptor 1 (*Tfr1*) was increased in whole-body *Ptpn2*-deficient mouse duodenal IECs vs. *Ptpn2*-WT or Het duodenal IECs (Figure 6B,D), indicating cellular iron deficiency [26]. Next, as systemic iron homeostasis revolves around the regulation of the iron exporter FPN1 by the iron regulatory hormone HAMP1 [8], we evaluated their expression levels. Surprisingly, expression of *Fpn1* and the ferroxidase hephaestin (*Heph*) were unaltered, and immunoblotting confirmed that the levels of the iron exporter were unchanged in duodenal IECs of *Ptpn2*-KO mice (Figure 7A–C). Interestingly, serum hepcidin levels were elevated in *Ptpn2*-deficient mice, a feature that is frequently present in anemia of chronic disease (ACD). The increase in serum hepcidin occurred despite reduced mRNA expression of the *Hamp1* gene, and no alteration of hepcidin protein levels in both liver and duodenal whole tissue lysates (WTLs) (Figure 7D–H) [27]. Gene expression of duodenal IEC-produced *Hamp1* was also unchanged (Figure 7I). Together, these data demonstrate that constitutive *Ptpn2* deficiency in mice results in alterations of iron handling genes in duodenal enterocytes and serum hepcidin concentrations, which may contribute to the iron deficiency and the manifestation of anemia.

## 3. Discussion

Proteomic analyses of serum samples from *PTPN2*-genotyped IBD patients identified the “iron homeostasis signaling pathway” as the most significantly downregulated pathway in CD patients with the rs1893217 risk allele. A major feature of these analyses was that the influence of the *PTPN2* variant on iron handling proteins differed considerably compared to the influence of disease severity in respect to the suite of proteins affected, the directionality of their expression, and the IBD subtype involved. Collectively, these data suggest that the influence of PTPN2 on iron homeostasis was, at least in part, distinct from the effects of inflammation. We confirmed that constitutive whole-body *Ptpn2*-KO mice exhibit features of anemia, including reduced hemoglobin concentrations accompanied by serum and tissue iron deficiency. Absorption of ^55^Fe through oral gavage was significantly diminished in *Ptpn2*-deficient mice, thereby identifying a primary defect in iron absorption. In support of these findings, the expression of the major iron transporter DMT1 in the duodenal brush border was dramatically reduced, thus providing a likely mechanism by which iron absorption was compromised in *Ptpn2*-KO mice.

While anemia in IBD patients has been extensively examined along with hematological and iron indices that determine the type of anemia, our understanding of how anemia occurs in IBD, and how disease-relevant genetic variants contribute to its development, is poorly developed [28]. In our study, we observed that serum levels of the iron carrier protein TF were reduced in IBD patients carrying the *PTPN2* risk allele, while soluble TFR1 was only reduced in CD patients (Figure 1), indicating that circulatory iron handling is compromised. The effects on TF may be expected, as iron stores are normally diminished in IBD patients and any iron present in the serum is immediately trafficked to tissues [2]. The reduced expression of serum TF is also frequently seen in IBD patients; however, serum-soluble TFR1 is not affected by chronic inflammation, though inflammatory events can indirectly affect TFR1 levels by inhibiting steady state erythropoiesis [29,30,31]. Serum levels of soluble TFR1 also inversely reflect the amount of iron available for erythropoiesis, indicating that CD patients with the rs1893217 risk allele may have functional iron deficiency, a condition featuring adequate iron stores but limited iron accessibility for erythrocyte precursors [30]. Functional iron deficiency is largely associated with accumulation and retention of iron stores in cells, including pro-inflammatory (M1) macrophages [32]. The increase in iron levels can be accompanied by production of reactive oxygen species (ROS), which can lead to lipid peroxidation via the Fenton reaction and ultimately ferroptosis [33]. Our group has shown that *PTPN2*-deficient macrophages preferentially differentiate into M1 macrophages; however, we have yet to determine whether they have increased iron stores and display an iron retention phenotype [34].

Our analysis also revealed downregulation of the serum heme scavenger HPX in UC patients, yet upregulation was seen in CD patients with the *PTPN2* risk allele (Figure 1A,B), suggesting differential regulation of this protein between the IBD subtypes; however, studies have not fully investigated the expression of iron handling proteins in CD patients compared to UC patients. Notably, the prevalence of anemia in CD patients has been suggested to be higher than UC patients [2]. Increased expression of the hemoglobin-binding proteins HP and HPR selectively in UC patients could be a potential mechanism to limit iron loss (Figure 1B). Serum levels of HP and HPR can be dramatically induced by various pro-inflammatory cytokines during an inflammatory response [35]. Given that PTPN2 is a critical negative regulator of pro-inflammatory signaling cascades, impaired activity may contribute to the elevation of serum HP and HPR levels [36,37]. In contrast, HPX is not considered an acute phase reactant in humans, thus the reduced HPX levels in UC patients could possibly reflect the recent extracellular release of heme compounds [38].

Notably, several serum iron handling proteins were altered by disease severity. The major serum hepcidin carrier protein A2M was downregulated in CD patients, suggesting hepcidin levels are dysregulated [39]. Interestingly, the hemoglobin-binding protein HP was also downregulated in CD patients. Downregulation of RAB7A, a regulator of endolysosomal organelle iron content, was seen in both CD and UC patients [40]. APOE, which has been shown to be required for tissue iron homeostasis, was downregulated in UC patients [41]. These findings demonstrate a distinction in the regulation of iron handling proteins between the *PTPN2* risk allele and severity of disease.

In the *Ptpn2*-KO mouse model, the significantly reduced serum iron levels and transferrin saturation are indicative of iron deficiency (Figure 2A,B) [17]. Additionally, *Ptpn2*-deficient mice have reduced hematocrit levels and hemoglobin concentrations—detected by CBC analyses—thus supporting the presence of anemia (Figure 2C,D) [18,19]. In differentiating the type of anemia, the mean corpuscular volume (MCV) (Figure 2E) of erythrocytes was reduced in *Ptpn2*-KO mice, consistent with iron deficiency anemia [19]. With these mice having iron deficiency anemia in the absence of chronic blood loss as determined by our hemoccult analysis, it was essential to investigate whether iron homeostasis was disturbed globally or only in select organs.

Non-heme iron analyses in *Ptpn2*-deficient mice revealed significantly reduced iron levels in major iron handling organs (Figure 3A–D), demonstrating that body iron stores are potentially depleted. Furthermore, *Ptpn2*-KO mice have profound splenomegaly, which is correlated with the severity of iron deficiency anemia [42]. The presence of splenomegaly is likely due to an increase in erythrocyte recycling and erythropoiesis as a consequence of anemia [43]. Although we did not assess heme iron levels in cardiac or gastrocnemius muscle, where myoglobin iron is abundantly located, non-heme iron content in the liver is the main indicator of body iron levels [8,25]. In addition to these findings, we found that *Ptpn2*-deficient mice have reduced non-heme iron levels in the duodenum and jejunum (Figure 3E,F), further suggesting that proximal small intestinal iron homeostasis is compromised.

Intestinal absorption of non-heme iron is the primary form of dietary iron for mice as they are poor heme absorbers [44]. Non-heme iron is also the main form of iron available in their diets. Non-heme iron absorption is driven by the actions of DMT1, which is energized by the H^+^-electrochemical potential gradient provided by NHE3 and basolateral Na^+^-K^+^-ATPase activity [45,46]. In this study, we demonstrated that the ^55^Fe content in the duodenal IECs and liver of *Ptpn2*-KO mice was greatly depleted (Figure 4A,B). This finding was supported by the reduced apical membrane expression of DMT1 (Figure 5F), indicating an inability to absorb iron at the duodenal brush border. We also expected that reduction of luminal Fe^3+^ to Fe^2+^ is not required, as we supplemented the gavage solution with ascorbic acid [24,25]. Moreover, the iron deficiency present in duodenal IECs, demonstrated by the reduced expression of the iron storage protein FT and increased expression of the basolateral iron importer TFR1, is likely due to impaired iron absorption (Figure 6C,D) [26]. Interestingly, serum hepcidin levels were elevated in *Ptpn2*-deficient mice despite reduced liver gene expression of *Hamp1*, unaltered hormone levels in the liver and duodenum, and unaltered protein levels of FPN1 in duodenal IECs (Figure 7A–I). A possible explanation could be a defect in the clearance of hepcidin, which is rapidly processed by the kidney. This may be due to some form of kidney dysfunction, as *Ptpn2*-deficient mice develop nephritis and the suppression of liver hepcidin expression is likely due to mechanisms related to iron deficiency [46,47,48]. Additionally, studies have shown that IBD patients can exhibit renal manifestations, including nephrolithiasis, tubulointerstitial nephritis, secondary (AA) amyloidosis, and glomerulonephritis, the latter of which involves improper filtration of toxins, metabolic waste, and excess fluid into the urine [49]. The increase in serum hepcidin levels may also be influenced by the inflammation present in *Ptpn2*-deficient mice [22]. Upregulation of hepcidin expression can occur through the JAK/STAT3 pathway by inflammatory cytokines including IL-6; however, we did not observe an alteration in serum levels of IL-6, although intestinal tissue levels of IL-6 were significantly upregulated in *Ptpn2*-KO mice [8,11,22]. Further investigations of serum creatinine and urine albumin levels will determine if kidney function is impaired [49]. Of note, the increased serum hepcidin levels are indicative of anemia of chronic disease, a condition that is commonly associated with inflammatory disorders [47]. This finding, collectively with our data demonstrating that *Ptpn2*-deficient mice have iron deficiency anemia, is consistent with the anemia found in the majority of IBD patients [50].

Although our data suggest impaired iron absorption is associated with reduced expression of DMT1, we have yet to identify the mechanism whereby the loss of *Ptpn2* alters DMT1 protein levels. DMT1 expression is regulated at multiple levels via transcriptional, translational, and post-translational mechanisms [8,46]. It is unlikely that the elevated serum hepcidin levels seen in *Ptpn2*-deficient mice contribute to the reduction in DMT1 expression by the iron-responsive element–iron regulatory protein (IRE/IRP) system, as the duodenal IECs in *Ptpn2*-KO mice are iron-deficient and ferroportin levels are unaltered [8,46]. Interestingly, the adapter protein, Nedd4 family-interacting protein 1 (NDFIP1), has been shown to regulate DMT1 by ubiquitination [51,52]. The pro-inflammatory cytokine tumor necrosis factor-alpha (TNF-α) has been shown to reduce expression of DMT1 in vitro as well [53,54]. Moreover, serum TNF-α concentrations are increased in *Ptpn2*-deficient mice [22]. Subsequent studies will determine whether NDFIP1 or TNF-α can alter DMT1 expression in IECs with loss of *Ptpn2*. These findings suggest a novel association between anemia, a clinically relevant and relatively common blood disorder, as well as a major extraintestinal manifestation of chronic inflammatory disease, and loss-of-function of the IBD-risk gene, *PTPN2*.

## 4. Materials and Methods

### 4.1. Swiss IBD Cohort Patient Sample Analysis

Active or quiescent disease in CD and UC patients (48% female) expressing the *PTPN2* wild-type (TT), heterozygous (CT), or homozygous LOF rs1893217 SNP (CC) alleles was determined by histologic assessment and Montreal classification (Table 1A–C). Two-dimensional gel electrophoresis followed by liquid chromatography with tandem mass spectrometry proteomics analysis was performed at the City of Hope Translational Biomarker Discovery Core Facility (Duarte, CA, USA) to screen for altered proteins in serum samples from *PTPN2*-genotyped IBD patients. Proteins with a corrected fold change of >1.5 were considered to be significantly altered and included in pathway analysis software (DAVID v6.7 and Ingenuity Pathway Analysis (IPA) 470319M) to identify biological pathways altered by one or more copies of the *PTPN2* risk ‘C’ allele.

### 4.2. Animals

#### 4.2.1. Animal Experiments

All animal procedures were performed according to institutional guidelines and approved by the University of California, Riverside Institutional Animal Care and Use Committee. Mice were held in a specific-pathogen-free facility with unrestricted access to food and water.

#### 4.2.2. Mouse Housing and Husbandry

BALB/c mice with constitutive *Ptpn2* deficiency were generated by breeding heterozygous (Het) mice as previously described [20,48]. All mice used for analyses were approximately 18–21 days old, as *Ptpn2*-knockout (KO) mice develop systemic inflammation at 3–4 weeks of age and succumb by 5 weeks of age [20,48]. Mice were fed a standard laboratory rodent diet containing 184 ppm iron (PicoLab^®^ Rodent Diet 20, Purina LabDiet; Richmond, IN, USA).

#### 4.2.3. Blood and Tissue Analyses

Automated complete blood count (CBC) was performed with the Hemavet 950FS at the University of California, Davis Comparative Pathology Laboratory (Davis, CA, USA). Serum iron (SI) and unsaturated iron-binding capacity (UIBC) were analyzed using the Iron-SL and UIBC kits (Sekisui Diagnostics, Burlington, MA, USA), according to the manufacturer’s guidelines. Transferrin (TF) saturation percentage was computed as SI/(SI + UIBC) × 100.

The non-heme iron content of the liver, spleen, kidney, gastrocnemius (skeletal) muscle, small intestine, and large intestine samples was analyzed by using an acid-digestion, chromogen-based colorimetric assay as described [55] and normalized by tissue weight.

#### 4.2.4. Duodenal Epithelial Cell Isolation

Proximal duodenal segments were harvested, cut open lengthwise, and incubated in ice-cold Cell Recovery Solution (354253, Corning; Corning, NY, USA) for 2 h. Duodenal tissue was then shaken to release the duodenal crypts and villi into solution. Samples were centrifuged at 2.5× *g* for 10 min at 4 °C. The supernatant was aspirated, and duodenal intestinal epithelial cells (IECs) were washed with ice-cold phosphate-buffered saline (PBS) (2×) and partitioned: 70% for protein and 30% for RNA. For protein, duodenal IECs were incubated with RIPA buffer (50 mM Tris-Cl pH 7.4, 150 mM NaCl, 1% NP-40, 0.5% sodium deoxycholate, and 0.1% SDS) supplemented with phosphatase inhibitor cocktails 2 and 3 (Sigma-Aldrich, St. Louis, MO, USA; P5726 and P0044, respectively), 200 mM sodium orthovanadate, protease inhibitor cocktail (Roche, Basel, Switzerland), 200 mM phenylmethylsulfonyl fluoride, and 200 mM sodium fluoride, and stored at −80 °C. For RNA, 350 uL of RLT buffer (74106; Qiagen, Hilden, Germany) and 10 µL β-mercaptoethanol were added to the duodenal IECs and stored at −80 °C.

### 4.3. qPCR

The total RNA from the duodenal IEC samples in RLT buffer and β-mercaptoethanol was extracted using the RNeasy Mini Kit (74106; Qiagen, Hilden, Germany) according to the manufacturer’s instructions. The RNA concentration was quantified by measuring the ratio of absorbance at 260/280 nm and 260/230 nm using a NanoDrop 2000c spectrophotometer (Thermo Scientific, Waltham, MA, USA). Complementary DNA (cDNA) synthesis was performed using the qScript cDNA SuperMix (95048; Quantabio, Beverly, MA, USA) following the manufacturer’s instructions. Real-time qPCR was performed using iQ SYBR Green Supermix (1708882; Bio-Rad, Hercules, CA, USA) on a C1000 Thermal cycler (Bio-Rad, Hercules, CA, USA) with a CFX96 Real-Time PCR system using the Bio-Rad CFX Manager 3.1 software according to the manufacturer’s guidelines. All primers were designed using NCBI Primer-BLAST, and the primer specificity and optimal annealing temperature were determined empirically. The real-time qPCR cycling protocol contained an initial denaturation and enzyme activation step (3 min, 95 °C), followed by 40 cycles of denaturing (95 °C, 10 s), annealing (55–64 °C, 10 s), and extension (72 °C, 10 s) steps. The results were analyzed by the 2^ΔΔCT^ method, and data graphs show geometric means with geometric standard deviations in a log10 scale. Primer information and reference genes (RG) are in Table 2.

### 4.4. Western Blot

Duodenal IEC samples in RIPA buffer supplemented with protease inhibitors were lysed using a sonicator (Q125; QSonica, Newtown, CT, USA) with the following settings: 30% amplitude for 40 s with 10-s intervals. Cell lysates were then centrifuged at 13,000 rpm for 10 min at 4 °C and the supernatant was recovered. Total protein abundance was estimated by the Pierce BCA assay (23225; Thermo Scientific, Waltham, MA, USA). Equivalent amounts of protein were loaded onto polyacrylamide gels, and after gel electrophoresis were transferred onto polyvinylidene difluoride membranes. Non-specific epitopes were blocked with 5–8% milk in 1× TBST (1× Tris-buffered saline, 0.1% Tween-20) for 1 h at room temperature. Membranes were incubated with primary antibody overnight at 4 °C, washed with 1× TBST, and incubated with their respective horseradish-peroxidase-conjugated secondary antibody for 1 h at room temperature. Proteins were detected with X-ray films (XAR ALF 2025; Lab Scientific Biokemix, Inc., Danvers, MA, USA) using the SuperSignal West Pico PLUS chemiluminescence detection kit (34580; Thermo Scientific, Waltham, MA, USA). Antibody information is in Table 3.

### 4.5. Imaging

Proximal duodenal segments were harvested and fixed in 4% paraformaldehyde overnight at 4 °C. Samples were then washed twice with 1× PBS and dehydrated with increasing concentrations of ethanol washes using a Shandon Excelsior ES Tissue Processor (Thermo Scientific, Waltham, MA, USA). Paraffin embedding was performed using Histoplast LP (Thermo Scientific, Waltham, MA, USA) in a Tissue-Tek station set (Miles Scientific, Newark, DE, USA). Paraffin blocks were sectioned in a rotary microtome (RM2235; Leica, Wetzlar, Germany) 5-μm thick, and placed on a charged microscope slide (1358W; Thermo Scientific, Waltham, MA, USA). Slides were deparaffinized and rehydrated before immunostaining.

Heat-induced antigen retrieval was performed for 20 min at approximately 95 °C, with the respective antigen retrieval buffer for each primary antibody and epitope interaction. Non-specific epitopes were blocked with blocking buffer (2% normal donkey serum (Jackson ImmunoResearch, West Grove, PA, USA), 1% bovine serum albumin (A9418; Sigma-Aldrich, St. Louis, MO, USA), 0.1% Triton-X (BP151; Thermo Scientific, Waltham, MA, USA), 0.05% Tween-20 (BP337; Thermo Scientific, Waltham, MA, USA), and 0.05% sodium azide (S8032; Sigma-Aldrich, St. Louis, MO, USA) in 1× PBS) for 30 min at room temperature. Primary antibodies were incubated for 1 h at room temperature. Detection was performed with the biotin–streptavidin system. Prolong Gold with 4′,6-diamidino-2-phenylindole (DAPI) was added to the samples according to the manufacturer’s instructions (P36936; Invitrogen, Carlsbad, CA, USA). Confocal images were obtained on an inverted Zeiss 880 microscope (Oberkochen, Germany). The antibody information, biotin–streptavidin conditions, and antigen retrieval methods are shown in Table 4.

To measure the abundance of NHE3 and DMT1 proteins on the apical surface of duodenal enterocytes, the brush border surface of duodenal IECs was traced with a fixed region of interest (ROI) that included the apical plasma membrane and subapical zone (~3 µm) in ImageJ 1.53o [56,57,58]. This area was considered the apical compartment where functional (plasma membrane) and reserve (sub-apical recycling endosome) transport carriers are expected to reside and traffic [56,57,58]. Each duodenal villus (7–10 per sample) was measured with the ROI in five representative locations and the mean fluorescence intensity (MFI) of all villi was averaged. Background subtraction was normalized through brightness and contrast of the immunofluorescent (IF) images.

### 4.6. Enzyme-Linked Immunosorbent Assay (ELISA)

Hepcidin ELISA for mouse serum (LS-F24250), liver (LS-F11620), and duodenum (LS-F36406) whole tissue lysates were obtained from LSBio (Seattle, WA, USA) and performed according to the manufacturer’s guidelines.

### 4.7. In Vivo Radiotracer Iron Assays

Conscious mice were administered ^55^Fe (1 µCi per gram of body weight as FeCl_3_ in 1 mM NaCl, 1 mM L-ascorbic acid, and 1 mM L-glutamine in 1× PBS) via oral-intragastric gavage following an overnight fast [24,25]. Mice were then euthanized at 2 h, and liver samples were collected [25]. Duodenal IECs were collected into 1.5 mL microcentrifuge tubes as described above. Half of the sample was used to quantify ^55^Fe and the remaining amount was used to measure the protein concentration by the Pierce BCA assay [24,25]. Tissue samples in SOLVABLE (Revvity, Inc., Waltham, MA, USA) were further processed according to the manufacturer’s protocol. The Hionic-Fluor (Revvity, Inc., Waltham, MA, USA) cocktail was added, and sample radioactivity was measured by using liquid scintillation counting (Beckman Coulter LS6500). ^55^Fe (specific activity: 1.0 mCi/mg) was obtained from the National Isotope Development Center (Oak Ridge, TN, USA).

### 4.8. Statistical Analyses

Data sets were analyzed through GraphPad Prism 10 and presented as mean ± standard deviation. Normality (Shapiro–Wilk) and outlier removal (ROUT), with a false discovery rate of 1% of the data, were performed. Statistical analysis was performed either by one-way analysis of variance (ANOVA) and Dunnett’s post-hoc analysis or Student’s *t*-test, with a critical significance level set at α = 0.05.

## 5. Conclusions

In conclusion, our study demonstrates a novel role of PTPN2 in systemic iron homeostasis. More importantly, our findings support a possible novel mechanistic regulation of iron handling proteins, including the major iron transporter DMT1, by PTPN2 in mice. Given that *PTPN2* SNPs are associated with several autoimmune disorders that can manifest dysregulated iron homeostasis, investigating the functional effects of *PTPN2* loss on iron handling proteins will provide further insight into mechanisms leading to iron deficiency in chronic disease and ultimately inform the generation of novel therapeutics for patients with iron deficiency [59,60].

## Figures and Tables

**Figure 1 ijms-26-03356-f001:**
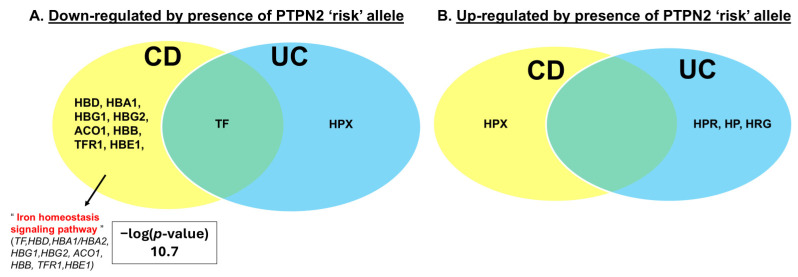
*PTPN2* regulates serum iron handling proteins in IBD patients. Two-dimensional gel electrophoresis was followed by liquid chromatography, along with tandem mass spectrometry proteomics analysis of serum samples from *PTPN2*-genotyped IBD patients. Serum proteins with an adjusted fold change of >1.5 were considered to be significantly altered and included in the DAVID v6.7 and IPA pathway analysis software 470319M. Serum iron proteins were altered in CD and UC patients (**A**,**B**) with the presence of the *PTPN2* rs1893217 risk allele independent of disease activity. Serum iron proteins were altered in CD and UC patients (**C**,**D**) by disease severity. n = 10 for each genotype and IBD subtype; 60 patient samples total.

**Figure 2 ijms-26-03356-f002:**
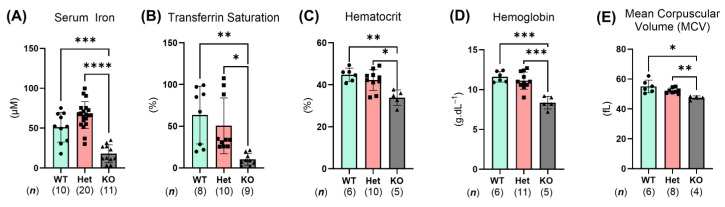
Constitutive *Ptpn2*-deficient mice show features of anemia, including iron deficiency. Blood iron and hematological parameters in *Ptpn2*-WT, Het, and KO mice: serum iron (**A**), % of serum transferrin saturated with iron (**B**), hematocrit % (**C**), hemoglobin concentrations (**D**), and mean corpuscular volume (MCV) of erythrocytes (**E**). Columns display means ± SD. * *p* < 0.05, ** *p* < 0.005, *** *p* < 0.0005, **** *p* < 0.0001.

**Figure 3 ijms-26-03356-f003:**
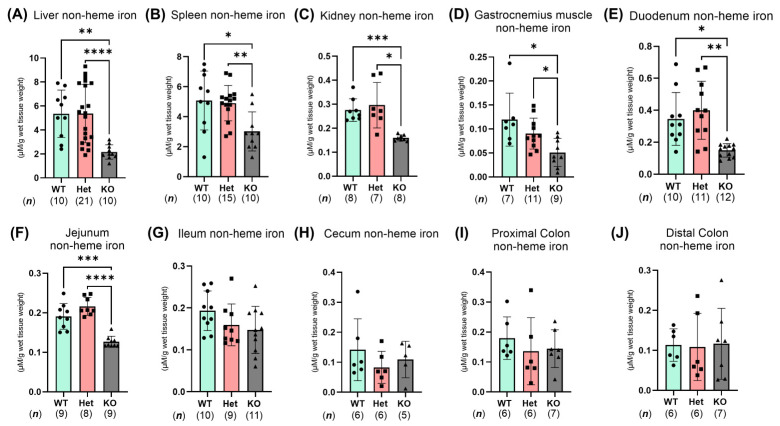
*Ptpn2*-KO mice have reduced non-heme iron levels in extraintestinal tissues and impaired iron homeostasis in the small intestine. Non-heme iron analysis of several extraintestinal tissues: liver (**A**), spleen (**B**), kidney (**C**), and gastrocnemius (skeletal) muscle (**D**) and of intestinal tissues: duodenum (**E**), jejunum (**F**), ileum (**G**), cecum (**H**), proximal colon (**I**), and distal colon (**J**). Columns display means ± SD. * *p* < 0.05, ** *p* < 0.005, *** *p* < 0.0005, **** *p* < 0.0001.

**Figure 4 ijms-26-03356-f004:**
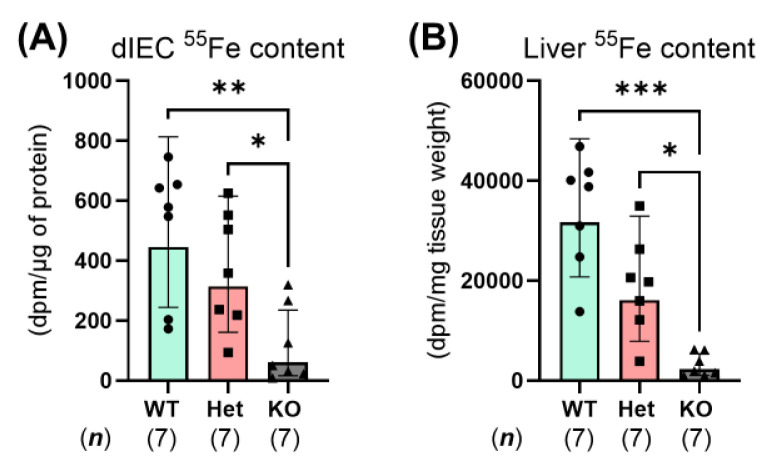
*Ptpn2*-KO mice have reduced iron absorption. Conscious *Ptpn2*-WT, Het, and KO mice were dosed with 1 µCi ^55^Fe per gram body weight via oral-intragastric gavage following an overnight fast. ^55^Fe content in duodenal IECs (dIECs) (**A**) and liver (**B**) of *Ptpn2*-WT, Het, and KO mice at 2 h post oral gavage. Columns display means ± SD. * *p* < 0.05, ** *p <* 0.005, *** *p <* 0.0005.

**Figure 5 ijms-26-03356-f005:**
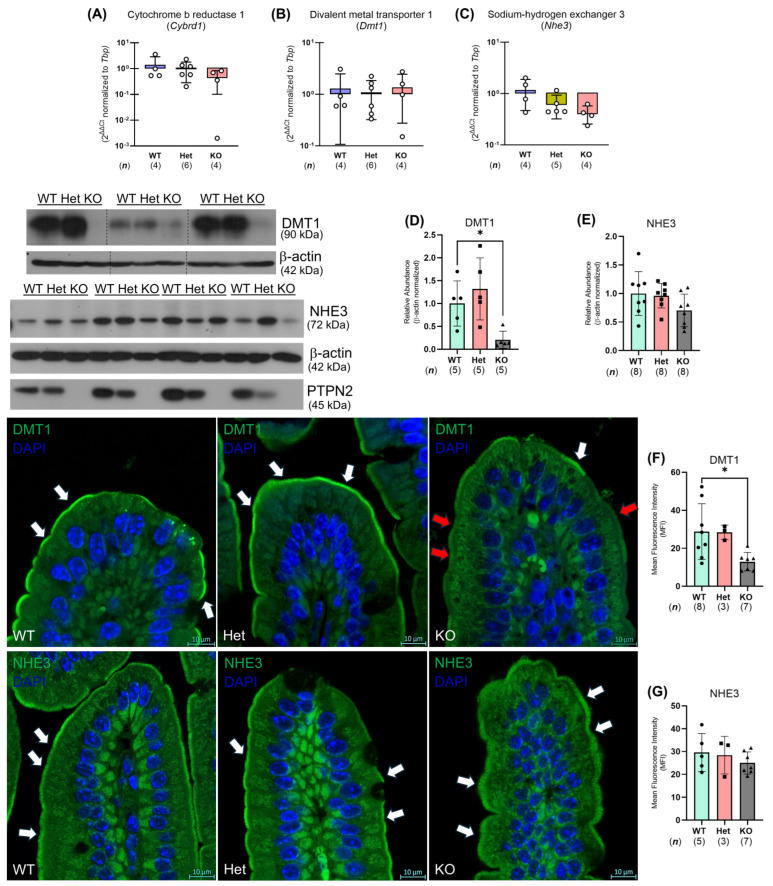
Duodenal epithelium of whole-body *Ptpn2*-KO mice has reduced expression of the iron transporter DMT1. Quantitative polymerase chain reaction (qPCR) analyses of iron uptake genes: the intestinal ferrireductase cytochrome b reductase 1 (*Cybrd1*) (**A**), the brush border ferrous iron transporter divalent metal transport 1 (*Dmt1*) (**B**), and the iron absorption driver sodium-hydrogen exchanger 3 (*Nhe3*) (**C**) of duodenal IECs from *Ptpn2*-WT, Het, and KO mice. Western blots and densitometry analysis of DMT1 (**D**) and NHE3 (**E**) protein levels in duodenal IECs from *Ptpn2*-WT, Het, and KO mice. Immunohistochemistry images of duodenal sections stained for DMT1 (**F**) and NHE3 (**G**) in *Ptpn2*-WT, Het, and KO mice. White arrows indicate staining of the apical membrane, while red arrows indicate reduced staining of the apical membrane of the duodenal epithelium. Mean fluorescence intensity of the staining was determined with ImageJ 1.53o. Columns display means ± SD. * *p <* 0.05.

**Figure 6 ijms-26-03356-f006:**
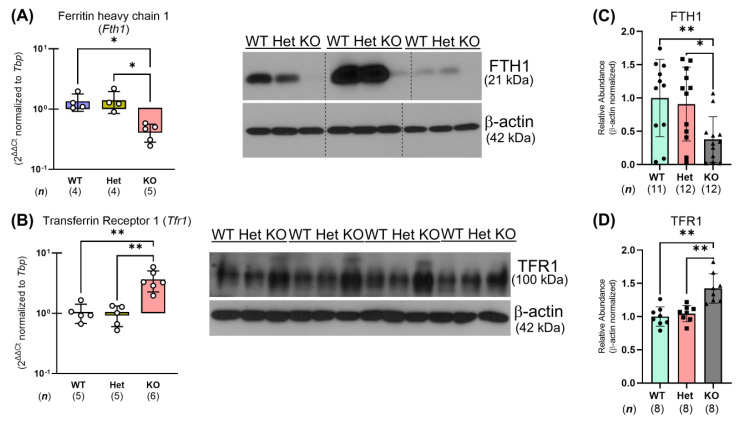
*Ptpn2*-KO mouse duodenal IECs are iron deficient. Gene expression of the iron storage molecule ferritin (*Ft*) through ferritin heavy chain 1 (*Fth1*) (**A**) and the basolateral iron importer transferrin receptor 1 (*Tfr1*) (**B**) in isolated duodenal IECs from *Ptpn2*-WT, Het, and KO mice. Western blots and densitometry analysis of FTH1 (**C**) and TFR1 (**D**). Columns display means ± SD. * *p <* 0.05, ** *p <* 0.005.

**Figure 7 ijms-26-03356-f007:**
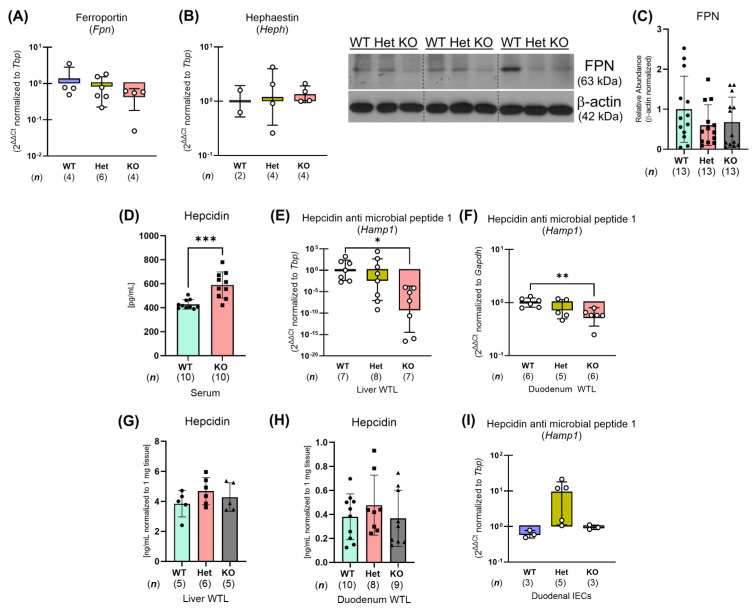
*Ptpn2*-deficient mice have elevated serum hepcidin levels. Iron exporter ferroportin (*Fpn*) (**A**) and the intestinal ferroxidase hephaestin (*Heph*) (**B**) expression assessed through qPCR analysis. Western blots and densitometry analysis of FPN levels (**C**). Serum hepcidin concentrations present in *Ptpn2*-WT, Het, and KO mice, measured by ELISA (**D**). Gene expression of hepcidin anti-microbial peptide 1 (*Hamp1*) in liver (**E**) and duodenal (**F**) whole tissue lysates (WTLs). ELISAs of hepcidin concentrations in liver (**G**) and duodenal (**H**) WTLs of *Ptpn2*-WT, Het, and KO mice. qPCR analysis of duodenal IEC *Hamp1* (**I**). Columns display means ± SD. * *p <* 0.05, ** *p <* 0.005, *** *p <* 0.0005.

**Table 1 ijms-26-03356-t001:** (**A**)**.**
*PTPN2* genotype and disease activity of IBD patients. CD and UC patients (48% female) expressing the *PTPN2* wild-type (TT), heterozygous (CT), or homozygous rs1893217 SNP (CC) alleles with active or quiescent disease as evaluated by histological assessment and Montreal classification. N = 10 for each genotype and IBD subtype; 60 patients total. (**B**)**.** CD patient characteristics. (**C**). UC patient characteristics.

(**A**)
	**Crohn’s Disease**	
	**Active**	**Quiescent**	**Total**
**CC allele**	3	7	10
**CT allele (Het)**	6	4	10
**TT allele (WT)**	6	4	10
**Sub-total**	15	15	30
	**Ulcerative Colitis**	
	**Active**	**Quiescent**	**Total**
**CC allele**	2	8	10
**CT allele (Het)**	6	4	10
**TT allele (WT)**	6	4	10
** *Sub-total* **	14	16	30
(**B**)
**Variant**	**Diagnosis**	**Severity**	**Sex**	**Age**	**Montreal Classification**
AA/TT	CD	Severe	M	66	B2p
AA/TT	CD	Severe	F	29	B1
AA/TT	CD	Severe	F	20	B2
AA/TT	CD	Severe	M	21	B2p
AA/TT	CD	Severe	F	25	B1
AA/TT	CD	Moderate	F	40	B3
AA/TT	CD	Quiescent	F	66	B1
AA/TT	CD	Quiescent	M	55	B1
AA/TT	CD	Quiescent	F	34	B2
AA/TT	CD	Quiescent	F	16	B1
GA/CT	CD	Severe	F	24	B1
GA/CT	CD	Moderate	F	35	B2p
GA/CT	CD	Moderate	F	60	B1
GA/CT	CD	Moderate	M	36	B1p
GA/CT	CD	Moderate	F	31	B2
GA/CT	CD	Mild	M	42	B2
GA/CT	CD	Quiescent	F	50	B1
GA/CT	CD	Quiescent	M	40	B1
GA/CT	CD	Quiescent	F	62	B1
GA/CT	CD	Quiescent	F	67	B2p
GG/CC	CD	Severe	M	67	B2
GG/CC	CD	Moderate	F	47	B2
GG/CC	CD	Moderate	F	41	B1p
GG/CC	CD	Quiescent	M	34	B1p
GG/CC	CD	Quiescent	M	25	B1p
GG/CC	CD	Quiescent	M	76	B1
GG/CC	CD	Quiescent	F	42	B1p
GG/CC	CD	Quiescent	M	70	B2
GG/CC	CD	Quiescent	F	13	B1
GG/CC	CD	Quiescent	M	14	B1
(**C**)
**Variant**	**Diagnosis**	**Severity**	**Sex**	**Age**
AA/TT	UC	Severe	M	31
AA/TT	UC	Severe	M	30
AA/TT	UC	Severe	M	54
AA/TT	UC	Severe	F	20
AA/TT	UC	Moderate	M	35
AA/TT	UC	Moderate	M	40
AA/TT	UC	Quiescent	M	47
AA/TT	UC	Quiescent	F	15
AA/TT	UC	Quiescent	F	26
AA/TT	UC	Quiescent	M	50
GA/CT	UC	Moderate	F	26
GA/CT	UC	Moderate	M	18
GA/CT	UC	Moderate	F	53
GA/CT	UC	Moderate	M	32
GA/CT	UC	Mild	F	52
GA/CT	UC	Mild	M	51
GA/CT	UC	Quiescent	F	45
GA/CT	UC	Quiescent	M	57
GA/CT	UC	Quiescent	M	25
GA/CT	UC	Quiescent	M	13
GG/CC	UC	Moderate	M	47
GG/CC	UC	Moderate	M	43
GG/CC	UC	Quiescent	F	10
GG/CC	UC	Quiescent	M	50
GG/CC	UC	Quiescent	F	48
GG/CC	UC	Quiescent	F	45
GG/CC	UC	Quiescent	M	52
GG/CC	UC	Quiescent	M	49
GG/CC	UC	Quiescent	M	31
GG/CC	UC	Quiescent	F	16

**Table 2 ijms-26-03356-t002:** qPCR primers.

Gene	Primer Sequence	GC%	Length	Tm	RefSeq
*Cybrd1*	F: CAGTGATTGCGACGGTTCTC	55%	20	56.1 °C	AF354666.1
	R: ATGGTACGAGGGGTGTTTCA	50%	20	56.0 °C	
*Dmt1*	F: CGCTCGGTAAGCATCTCGAA	55%	20	57.2 °C	NM_008732.2
	R: TGTTGCCACCGCTGGTATCT	55%	20	59.2 °C	
*Fpn*	F: CATTGCTAGAATCGGTCTT	42.1%	19	49.6 °C	NM_016917.2
	R: GCAACTGTGTCACCGTCAAAT	57.6%	21	56.1 °C	
*Fth1*	F: TGGAGTTGTATGCCTCCTACG	52.4%	21	56.3 °C	NM_010239.2
	R: TGGAGAAAGTATTTGGCAAAGTT	34.8%	23	53.0 °C	
*Hamp1*	F: AAGCAGGGCAGACATTGCGAT	52.4%	21	59.8 °C	AF503444.1
	R: CAGGATGTGGCTCTAGGCTATGT	52.2%	23	58.2 °C	
*Heph*	F:TAATCCCCGCCAGACAGGA	57.9%	19	58.1 °C	NM_001159627.1 and NM_181273.4
	R:GCCTTCAGCTTATTTACCTTGTT	39.1%	23	53.5 °C	
*Nhe3*	F: GAGGAGGAACCGAGCAGTGA	60%	20	58.8 °C	NM_001081060.2 and XM_006517022.5
	R: GTGGGACAGGTGAAAGACGATT	50%	22	57.1 °C	
*Tfr1*	F: TGGAATCCCAGCAGTTTCTT	45%	20	54.3 °C	NM_001357298.1
	R: GCTGCTGTACGAACCATTTG	50%	20	54.5 °C	
*Gapdh* (RG)	F: GTTTGTGATGGGTGTGAACCACG	52.2%	23	58.8 °C	NM_001289726.1
	R: GTGGCAGTGATGGCATGGAC	60%	20	59 °C	
*Tbp* (RG)	F: CCTTGTACCCTTCACCAATGAC	50%	22	55.6 °C	NM_013684.3
	R: ACAGCCAAGATTCACGGTAGA	47.6%	21	56.0 °C	

**Table 3 ijms-26-03356-t003:** Western blot antibodies.

Antibody	Source	Catalog #
DMT1	Dr. Francois Canonne-Hergaux (INSERM)	N/A
FPN	Thermofisher	PA5-22993
FTH1	Thermofisher	MA5-32244
NHE3	EMD Millipore (Burlington, MA, USA)	AB3085
PTPN2	Cell Signaling (Danvers, MA, USA)	58935
TFR1	Thermofisher	PA5-110480
β-actin	Sigma-Aldrich	A5316

**Table 4 ijms-26-03356-t004:** Immunohistochemistry guidelines.

Antibody	Source	Catalog #	Antigen Retrieval Buffer	Primary Concentration
DMT1	Dr. Francois Canonne-Hergaux (INSERM)	N/A	Tris-EGTA pH 9	1:100
NHE3	EMD Millipore	AB3085	Tris-EGTA pH 9	1:100

## Data Availability

Proteomic data will be uploaded to an appropriate publicly available repository upon publication, and made available upon request following publication of this manuscript.

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
