# Peer review of "PTPN2 Regulates Iron Handling Protein Expression in Inflammatory Bowel Disease Patients and Prevents Iron Deficiency in Mice"

_ijms, 2025, doi:10.3390/ijms26073356_

Round 1
Reviewer 1 Report
Comments and Suggestions for Authors
This study focuses on the role of PTPN2 in regulating iron metabolism in IBD patients, demonstrating clear clinical significance and scientific merit. The integration of clinical cohort analysis and animal models reveals the association between PTPN2 gene polymorphism and iron homeostasis, providing novel insights into anemia mechanisms in IBD.
Strengths and Weaknesses
(1) Innovation:
First to directly link PTPN2 gene polymorphism to iron metabolism dysregulation in IBD, identifying its regulation of DMT1 protein expression as a potential therapeutic target.
(2) Limitations:
Mechanistic insights into PTPN2-mediated DMT1 regulation (e.g., phosphorylation modification, protein interactions) require experimental validation (e.g., Co-IP, kinase activity assays).
The proposed "renal clearance defect" hypothesis for elevated hepcidin in Ptpn2-KO mice lacks direct evidence (e.g., kidney functional indices or hepcidin clearance rate measurements).
Major Concerns
(1) The conclusion claiming PTPN2 as "the master regulator of systemic iron homeostasis" requires caution. Analysis of interactions with other iron metabolism pathways (e.g., Hepcidin-FPN axis) should be included.
(2) Potential indirect effects of PTPN2 deficiency on iron metabolism via inflammation (e.g., IL-6/JAK-STAT3 pathway's role in hepcidin regulation) remain undiscussed.
(3) Clinical relevance could be strengthened by analyzing correlations between PTPN2 genotypes and iron therapy responses in IBD patients.
(4) The Discussion section should contrast findings with established PTPN2 immune regulatory functions and cite recent advances in IBD-related iron metabolism (e.g., 2023 Nat Immunol paper on macrophage ferroptosis).
Recommendations
(1)Supplement molecular mechanism studies: Validate PTPN2-DMT1 regulatory relationships through phosphorylation profiling and promoter binding assays.
(2)Enhance clinical translation: Analyze PTPN2 genotype associations with therapeutic outcomes of iron supplementation in IBD cohorts.
Reviewer 2 Report
Comments and Suggestions for Authors
Reviewer Comments:
In the paper “PTPN2 Regulates Iron Handling Protein Expression in Inflammatory Bowel Disease Patients and Prevents Iron Deficiency in Mice” by Lei et al., the authors have clearly explained the PTPN2 in regulating iron homeostasis. The experiment performed by the authors are clear and the data was explained very clearly. The methods and the result section is documented with utmost clarity. The minor issue is with the discussion section where I feel the author has explained the result section again, it looks very repetitive. The author should clearly discuss the results with putting some insight about the future study that can be carried out with the observation they got.
